# MULTI-TASK LEARNING ON MNIST IMAGE DATASETS

## ABSTRACT

We apply multi-task learning to image classification tasks on MNIST-like datasets. MNIST dataset has been referred to as the *drosophila* of machine learning and has been the testbed of many learning theories. The NotMNIST dataset and the FashionMNIST dataset have been created with the MNIST dataset as reference. In this work, we exploit these MNIST-like datasets for multi-task learning. The datasets are pooled together for learning the parameters of joint classification networks. Then the learned parameters are used as the initial parameters to retrain disjoint classification networks. The baseline recognition model are all-convolution neural networks. Without multi-task learning, the recognition accuracies for MNIST, NotMNIST and FashionMNIST are 99.56%, 97.22% and 94.32% respectively. With multi-task learning to pre-train the networks, the recognition accuracies are respectively 99.70%, 97.46% and 95.25%. The results re-affirm that multi-task learning framework, even with data with different genres, does lead to significant improvement.

## 1 INTRODUCTION

Multi-task learning (Caruana (1998)) enjoys the idea of pooling information that can be learned from data collected for multiple related tasks. Multiple sources of information can stem from multiple datasets, or even a single dataset, for multiple tasks. In this work, we focus on the case of using multiple datasets for multiple tasks. Namely, we use MNIST, FashionMNIST, and NotMNIST image datasets collected for digit recognition, fashion item recognition, and letter recognition, respectively.

Information sharing in multi-task training can be achieved in various formality. For neural-network based deep learning, the sharing can happen at the input layer, the hidden layers, or the output layer. Input-layer multi-tasking combines heterogeneous input data, hidden-layer multi-tasking shares multiple groups of hidden layer units, and output-layer multi-tasking pools multiple output groups of categories. The implementation of a multi-task learning system depends on the data and the tasks at hand.

Multi-task learning has been successfully applied to many applications of machine learning, from natural language processing (Collobert & Weston (2008)) and speech recognition (Deng et al. (2013)) to computer vision (Ren et al. (2015)) and drug discovery (Ramsundar et al. (2015)). A recent review of multi-task learning in deep learning can be found in (Ruder (2017)).

## 2 MNIST DATASETS

### 2.1 MNIST

The MNIST dataset (LeCun et al. (1998)) consists of a training set of 60,000 images, and a test set of 10,000 images. MNIST is often referred to as the *drosophila* of machine learning, as it is an ideal testbed for new machine learning theories or methods on real-world data. Table 1 lists the state-of-the-art performance on MNIST dataset. A few examples of MNIST dataset are shown in Table 2, alongside with examples of the other MNIST-like datasets.

Table 1: Recognition error rates on MNIST task of state-of-the-art systems

| Method | Error rate |
|---|---|
| Wan et al. (2013) | 0.21% |
| Ciregan et al. (2012) | 0.23% |
| Sato et al. (2015) | 0.23% |
| Chang & Chen (2015) | 0.24% |

## 2.2 FASHIONMNIST

Xiao et al. (2017) presents the FashionMNIST dataset. It consists of images from the assortment on Zalandos website. As given out by name, the configuration of the FashionMNIST dataset completely parallels the configuration of the MNIST dataset. FashionMNIST consists of a training set of 60,000 images and a test set of 10,000 images. Each image is a $28 \times 28$ grayscale image associated with a label from 10 classes. FashionMNIST poses a more challenging classification task than the MNIST digits data. A leaderboard for FashionMNIST has been created and maintained at `https://github.com/zalandoresearch/fashion-mnist`.

Table 2: Image examples of MNIST-like datasets

| MNIST | | FashionMNIST | | NotMNIST | |
|---|---|---|---|---|---|
| class | example | class | example | class | example |
| 0 | | T-Shirt/Top | | A | |
| 1 | | Trouser | | B | |
| 2 | | Pullover | | C | |
| 3 | | Dress | | D | |
| 4 | | Coat | | E | |
| 5 | | Sandals | | F | |
| 6 | | Shirt | | G | |
| 7 | | Sneaker | | H | |
| 8 | | Bag | | I | |
| 9 | | Ankle boots | | J | |

## 2.3 NOTMNIST

Bulatov (2011) presents the NotMIST dataset. NotMNIST dataset consists of more than 500k $28 \times 28$ greyscale images of English letters from $A$ to $J$, including a small hand-cleaned subset and a large uncleaned subset. From the uncleaned subset, we randomly select 60,000 examples as train set and 10,000 examples as test set.

## 3 METHOD

In short, we apply multi-task learning to learn from the MNIST-like datasets to pre-train the parameters of the all-convolution neural networks for individual image recognition tasks. The overall framework is depicted in Figure 1.

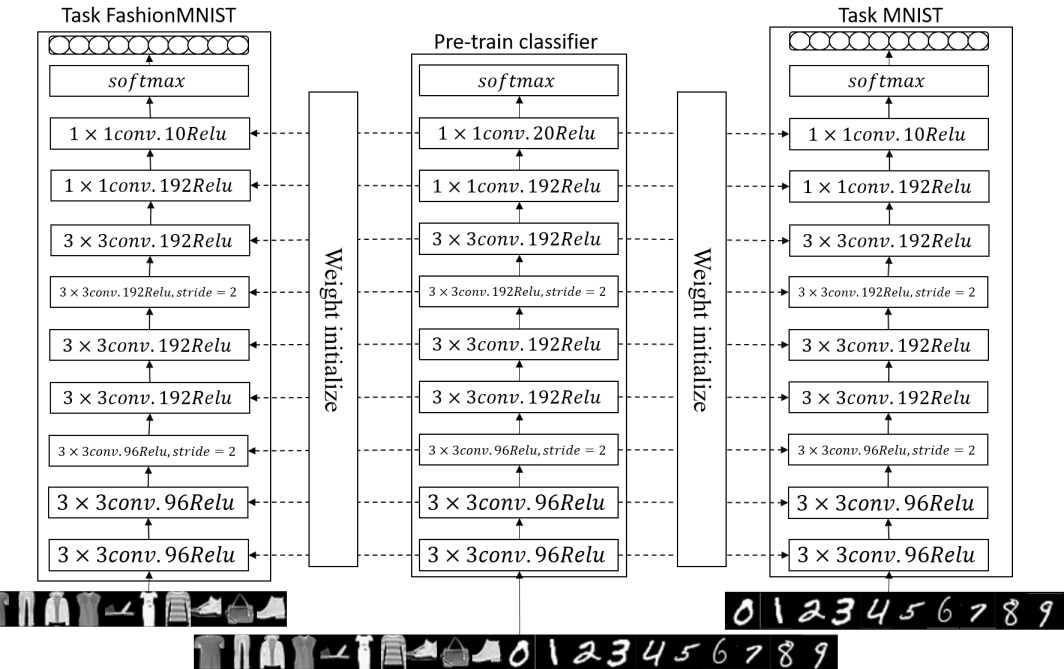

Figure 1: The proposed multi-task learning framework

### 3.1 ALL-CONVOLUTION NEURAL NETWORKS

A convolutional neural networks (CNN) (LeCun et al. (1998)) usually consists of alternating convolution layers and pooling layers. In contrast, an all-convolution neural network (Springenberg et al. (2014)) replace pooling layers by convolution layers. A feature map $\mathbf{f}$ created at a convolution layer can be represented by a three-dimensional tensor of order $W \times H \times N$, where $W$ and $H$ are the width and height, $N$ is the number of channels. A pooling operation with a size of $k \times k$ and a stride of $r$ applied to a three-dimensional tensor results in another tensor $\mathbf{s}$ with

$$s_{i,j,n} = \left( \sum_{w=-\lfloor k/2 \rfloor}^{\lfloor k/2 \rfloor} \sum_{h=-\lfloor k/2 \rfloor}^{\lfloor k/2 \rfloor} |f_{r \cdot i + w, r \cdot j + h, n}|^p \right)^{1/p} \tag{1}$$

where the first 2 subscripts index position in the map, and the third subscript indexed the channel. If a convolution operation with the same stride were applied to $\mathbf{f}$, we would have a tensor $\mathbf{c}$ with

$$c_{i,j,o} = \sigma \left( \sum_{w=-\lfloor k/2 \rfloor}^{\lfloor k/2 \rfloor} \sum_{h=-\lfloor k/2 \rfloor}^{\lfloor k/2 \rfloor} \sum_{n=1}^{N} t_{w,h,n,o} f_{r \cdot i + w, r \cdot j + h, n} \right) \tag{2}$$

where $\mathbf{t}$ is the convolution kernel tensor, and $\sigma(\cdot)$ is an activation function. Thus, a pooling operation can be seen as a convolution operation with uniform kernel tensor and with $L^p$-norm as the activation function.

## 4 EXPERIMENT

### 4.1 CLASSIFIER ARCHITECTURE AND MULTI-TASK LEARNING

The architecture of an all-convolution neural network for a single task is shown in Figure 2. The multi-task learning classifier has the same architecture as a single-task classifier except that the width of the output layer is proportional to the number of tasks. The target label is enhanced accordingly

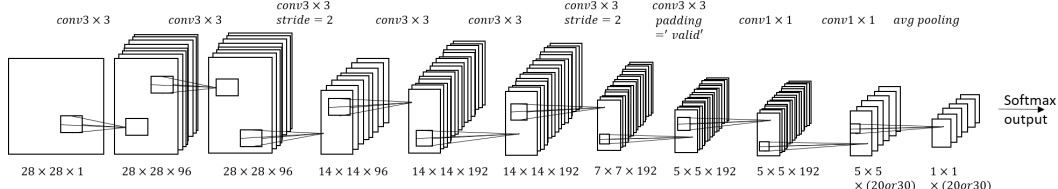

Figure 2: The architecture of an all-convolution neural network for a single task

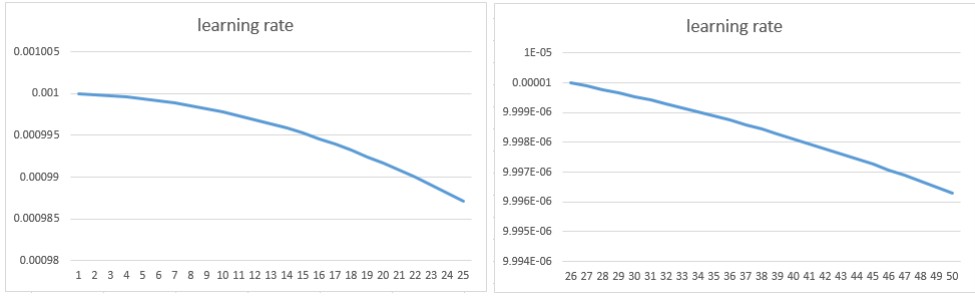

Figure 3: The architecture of all-convolution neural network for multi-task learning

by zero-padding. As mentioned before, we combine the training data together, so that the training label dimension is increased from $[1 \times 10]$ to $[1 \times 20]$ by zero-padding. This is shown in Figure 3.

Multi-task learning with different combination of datasets are evaluated, namely 3 bi-task learning and 1 tri-task learning. For example, we combine the MNIST train dataset and FashionMNIST train dataset together. Thus, there are 120,000 examples to the bi-task learning network for MNIST and FashionMNIST. Each training example has an input image size of $28 \times 28$ and an output 1-hot vector size of 20.

Each network is trained with 50 epochs. A two-stage learning rate decay scheme is implemented. The initial learning rate is $10^{-3}$ for the first stage of 25 epochs, and $10^{-5}$ for the second stage of 25 epochs. When training a model, it is often recommended to lower the learning rate as the training progresses. So, we let learning rate decay in each epochs. The decay rate is $\frac{1}{1+d \times n}$ where $d$ is the learning rate set in the begin divided by 25. $n$ is the number of current epochs. Figure 4 shows the learning rate scheme. The size of a mini-batch is set to 100, and the Adam optimizer is used.

When multi-task learning is complete, the parameters in the network is used to initialize single-task classifiers. Figure 5 illustrates how parameters are handed over. The single-task classifiers are then re-trained to perform their respective classification tasks.

Figure 4: The learning rate scheme.

## 4.2 RESULTS

The experimental results are summarized in Table 3. We are happy to see that multi-task learning works, even though the MNIST, NotMNIST, and FashionMNIST datasets are images from totally

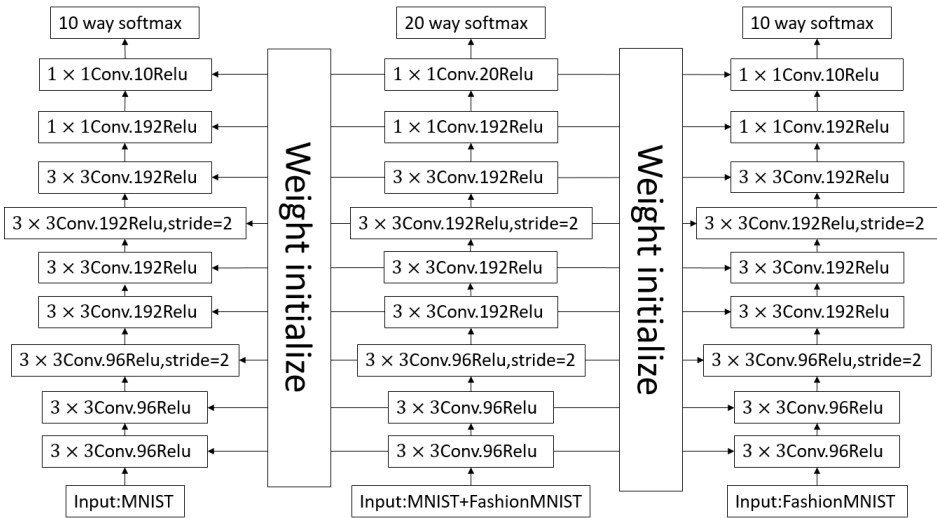

Figure 5: Passing parameters from multi-task learning network to single-task networks

Table 3: Image recognition accuracy rates for MNIST tasks with multi-task learning

| dataset | single-task | F+N | M+F | M+N | F+N+M |
|---|---|---|---|---|---|
| MNIST | 99.56% | – | **99.71%** | 99.70% | 99.70% |
| NotMNIST | 97.22% | 97.38% | – | 97.40% | **97.46%** |
| FashionMNIST | 94.32% | 95.20% | 95.18% | – | **95.25%** |

different classes. The bi-task learning systems are always better than the single-task systems. Furthermore, the tri-task learning systems are the best, except for the MNIST (0.01% difference). The relative reduction in error rates by tri-task learning are respectively 31.8% for MNIST (99.56% to 99.70%), 16.4% for FashionMNIST (94.32% to 95.25%), and 8.6% for NotMNIST (97.22% to 97.46%). The results confirm that multi-task learning is able to learn representation which is universal and robust to different tasks.

To better understand the effect of multi-task learning, we plot the distribution of the high dimensional-data of the classes by the $t$-distributed stochastic neighbor embedding (t-SNE) (van der Maaten & Hinton (2008)). Figure 6 shows the t-SNE of the output values of several hidden layers of the classifiers. The separation between the learned class manifolds appears to increase with the integration of multi-task learning. The representation learned with multi-task learning in the loop looks better indeed.

## 5 CONCLUSION

In this paper, we use multi-task learning in pre-training an all-convolution neural network model. We pass the parameters of trained multi-task models to single-task models. Evaluation on MNIST-like datasets show that using multi-task learning can improve image recognition accuracy. The more data we use, the better results we get. This agrees with statistical learning theory that using more data reduces the generalization gap, thus improving test set performance, even if the data comes from a different domain. The classification tasks of the images of digits, letters, and fashion items share parts of their hierarchical representations. By multi-task learning, it is possible to make such common representation robust to help individual classification tasks.

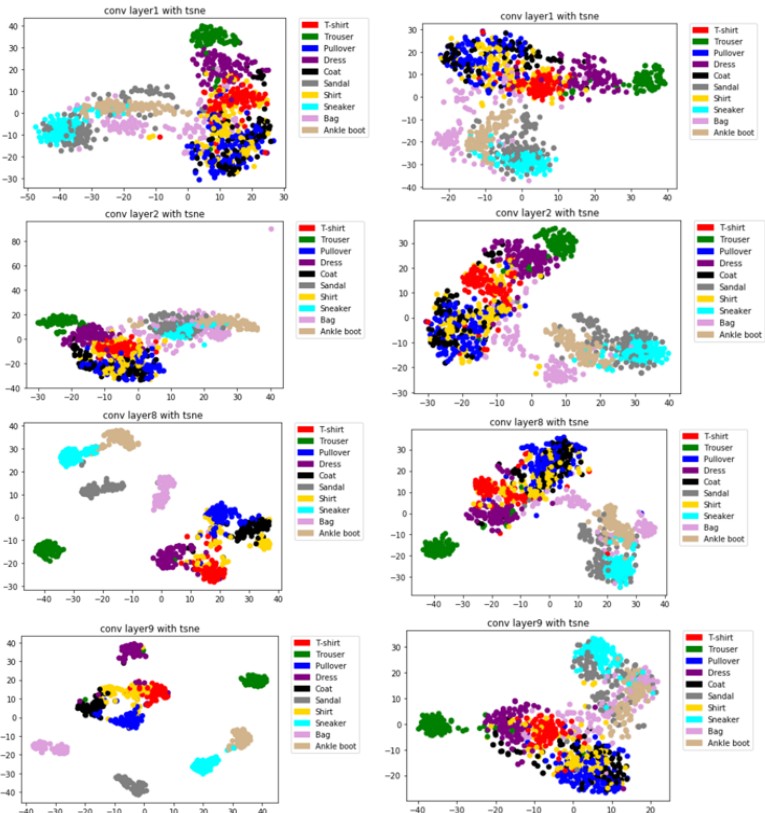

Figure 6: Visualization of data manifolds with t-SNE. The left column is the case with multi-task learning, and the right column is the case without multi-task learning.

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
