# OpenReview forum: "Multi-task Learning on MNIST Image Datasets"
_ICLR.cc/2018/Conference — Reject_

### Official Review · AnonReviewer3 · 2017-11-26
**a paper with limited novelty**

**Rating:** 5
**Confidence:** 4

**Review:**

This paper presents a multi-task neural network for classification on MNIST-like datasets.

The main concern is that the technical innovation is limited. It is well known that multi-task learning can lead to performance improvement on similar tasks/datasets. This does not need to be verified in MNIST-like datasets. The proposed multi-task model is to fine tune a pretrained model, which is already a standard approach for multi-task and transfer learning. So the novelty of this paper is very limited.

The experiments do not bring too much insights.

---

> ### Author Response · Authors · 2018-01-04
> **Response to ICLR 2018 Conference Paper318 AnonReviewer3**
>
> We thank the reviewer for the thorough analysis and insightful comments. Although it is well known that multi-task learning can lead to performance improvement on similar tasks, there is not too many research on FashionMNIST and NotMNIST datasets. And we also visualize the results obtained with/without multi-task learning to analyze our experiment result. Hopeful it contributes to the research community.

---

### Official Review · AnonReviewer1 · 2017-11-26
**The manuscript applies multi-task learning to the three MNIST-like datasets by simply pre-training the parameters of the networks using samples of datasets. The presented idea is quite simple and there is no technique contribution.**

**Rating:** 4
**Confidence:** 5

**Review:**

The manuscript mainly utilizing the data from all three MNIST-like datasets to pre-train the parameters of joint classification networks, and the pre-trained parameters are utilized to initialize the disjoint classification networks (of the three datasets).

The presented idea is quite simple and the authors only re-affirm that multi-task learning can lead to performance improvement by simultaneously leverage the information of multiple tasks. There is no technique contribution.

Pros:
1.	The main idea is clearly presented.
2.	It is interesting to visualize the results obtained with/without multi-task learning in Figure 6.

Cons:
1.	The contribution is quite limited since the authors only apply multi-task learning to the three MNIST-like datasets and there is no technique contribution.
2.	There is no difference between the architecture of the single-task learning network and multi-task learning network.
3.	Many unclear points, e.g., there is no description for “zero-padding” and why it can enhance target label. What is the “two-stage learning rate decay scheme” and why it is implemented? It is also unclear what can we observed from Figure 4.

---

> ### Author Response · Authors · 2018-01-04
> **Response to ICLR 2018 Conference Paper318 AnonReviewer1**
>
> Thank you for the very pertinent and exhaustive comments about our work.  We humbly accept the shortcomings you said about our paper and have already revised the shortcomings in our paper.
>
> About the unclear points, our answers as below:
> In order to compare the result between multi-task learning and single task, we must keep the architecture of the network unchanging.
> The reason we implement "zero-padding" is when the multi-task learning process is executed, we combine the training data of MNIST-like datasets together. In order to train a 20(or 30) ways classifier, we have to extend target labels by zero-padding.
> The “two-stage learning rate decay scheme” we implemented is just a way to decrease learning rate. We found it is helpful to improve recognition accuracies.

---

### Official Review · AnonReviewer2 · 2017-11-28
**The paper applies multi-task learning to MNIST type image datasets and gets reasonably interesting and expected results. However, there is not too much novelty in methodology or formalism in the work.**

**Rating:** 5
**Confidence:** 3

**Review:**

The paper applies multi-task learning to MNIST (M), FashionNIST (F), and NotMNIST (N) datasets. That is, the authors first train a neural network (with a specific architecture; in this case, it is an all-convolutional network) on a combination of the datasets (M+F; F+N; N+M; M+F+N) and then use the learned weights (in all but the output layer) to initialize the weights for task-specific training on each of the datasets. The authors observe that for each of the combinations, the above approach does better than training on a dataset individually. Further, in all but one case, initializing weights based on training on M+F+N gives the best performance. The improvements are not striking but are noticeable.

---

> ### Author Response · Authors · 2018-01-04
> **Response to ICLR 2018 Conference Paper318 AnonReviewer2**
>
> We thank the reviewer for the thorough analysis and insightful comments.

---

### Public Comment · (anonymous) · 2017-11-27
**Implementation details**

This is not really a comment. As we intend to reproduce the results in your paper, we would like to know more about the implementation details.

Could you please tell us:
1. if you used cross-entropy error or any other criteria as the loss function,
2. how many epochs did you use to train the pre-trained single-task models and what the learning rate and the mini-batch size were, and
3. what function was used to specify the decay of the learning rate for training the multi-task models?

Besides, would it be possible for you to share your code with us?

Thanks.

---

> ### Author Response · Authors · 2017-11-28
> **Dear Readers:**
>
> Thanks for your comments:
> 1.Yes, I use cross-entropy as the loss function in my experiment.
> 2.The hyper-parameters of single-task models are as same as multi-task models. Each network is trained with 50 epochs. A two-stage learning rate decay scheme is implemented. The initial learning rate is 0.001 for the first stage of 25 epochs, and 0.00001 for the second stage of 25 epochs. The size of a mini-batch is set to 100.
> 3.Learning rate decay function is: LearningRate = LearningRate * 1/(1 + decay * epoch).
> The decay argument is set to LearningRate /25
> 4.I will upload my code to github as soon as possible.
>
>
>
> Hope the response above clarified your questions.

---

> > ### Public Comment · (anonymous) · 2017-11-28
> > **Code for the Paper**
> >
> > Hello, we are also intending to reproduce the results of your paper. Could you please post the github link here as a comment once you manage to upload the code?
> >
> > Would you also be able to post the exact 60,000 train and 10,000 test data points you used from the Not MNIST data set, so that we could use the exact same datasets  to reproduce your results?

---

> > > ### Author Response · Authors · 2017-11-30
> > > **Code for the Paper**
> > >
> > > github link:https://github.com/st70712/image-multi-task-learning-

---

> > ### Public Comment · (anonymous) · 2017-12-01
> > **Design choices**
> >
> > Hi I am wondering why you chose the specific CNN structure (e.g. hyper-parameters) and why you chose this changing learning rate. Could you please justify your choices?

---

> > > ### Author Response · Authors · 2017-12-04
> > > **Design choices**
> > >
> > > Dear reader:
> > > The CNN model which we reference from 'Striving for Simplicity: The All Convolutional Net'.
> > > You can find more detail in their paper.The “two-stage learning rate decay scheme” we implemented is just a way to decrease learning rate.We found it is helpful to improve recognition accuracies.

---

### Public Comment · (anonymous) · 2017-12-12
**Details of the t-SNE plot**

Can you guys please comment on how did you generate the t-SNE plot or share the code for it.
Thanks

---

> ### Author Response · Authors · 2017-12-15
> **Details of the t-SNE plot**
>
> The code has been upload to Github:https://github.com/st70712/image-multi-task-learning-
> You can find the detail in "TSNE_CNN.ipynb" file.

---

### Public Comment · ~Amar_Kumar1 · 2017-12-16
**Reviews after reproducing the results**

This paper examined the improvement that multi-task learning could bring through training a fixed-structure
all-convolutional neuron network using three image text datasets. Multi-task learning was achieved by first training
a network in a pre-defined way to get the weights between layers and then using this weights to initialize the
networks for single-task learning.
1) Implementation: We tried to reproduce the accuracy results in the paper for single-task learning and multi-task
learning on MNIST, NotMNIST and FashionMNIST datasets. A same CNN structure was used throughout. The
authors provided the code which was simple, elegantly written and well formatted. Thus we were able to reproduce
the results very easily. For single-task learning, 50 epochs were used to train the model with a single dataset using
4:1 training and validation data split.
Few key differences were observed between the actual implemented model and the model presented in the
paper. Moreover, some important details regarding the model architecture were missing. For example, there was
no mention of batch normalization in the manuscript, although it was used in the code. Batch normalization was
used after each convolution layer. It was difficult to reproduce the results without such details. But as soon as the
code was given the entire structure was very clear. However, hyperparameters and structure of the model could
have been explained in more detail. Inclusion of some statistics comparing accuracy of single and multi-tasking
models would have been more useful.
2) Reproduced Results: Denoting MNIST, NotMNIST and FashionMNIST by M, N and F respectively, the
accuracies that we obtained on MNIST using M, M+N, M+F and M+N+F are 99.71%, 99.67%, 99.67% and
99.63% respectively; the results on NotMNIST for N, M+N, N+F and M+N+F are 97.55%, 97.53%, 97.43%,
97.54% and the results on FashionMNIST for F, F+N, M+F and M+N+F are 94.81%, 94.92%, 94.90% and 95.07
respectively.We obtained slightly different results as compared to manuscript. The reason behind this could be
different version of tensorflow (we used 1.4.0). It was very helpful of the authors to mention the exact version of
the libraries used which supported our ease of reproducing it.
When reproducing the results for single-task learning, we tried to tune the hyper-parameters and achieved a better
accuracy of 99.88% for single-task learning than in the paper. Hence, we believe that some more hyper-parameter
optimization and then implementing a multi-task learning would be helpful. Overall, this paper brings out a very
interesting insight which can be of great use if the same thing could be extended to other different domains

---

> ### Author Response · Authors · 2017-12-17
> **Reviews after reproducing the results**
>
> Thanks for your comments:
> I am interested in your fine tune of the hyper-parameters. How did you achieve 99.88% accuracy? Could you please tell me your hyper-parameters settings and network architecture?

---

### Public Comment · (anonymous) · 2017-12-16
**Reproducibility Challenge: Multi-task Learning on MNIST Image Datasets**

This paper seeks to apply multi task learning to MNIST-like image classification tasks. The authors intend to show that classifiers whose weights have been initialized from a multi task classifier trained on several different datasets will outperform single task classifiers trained on a single dataset pertaining to the single task.

The paper outline several methods of applying multi task learning to neural networks at the input layer, hidden layers, and final output layer. The experiment applies multi-task learning to recognizing digits, fashion items, and letters from the MNIST, FashionMNIST, and NotMNIST datasets respectively. It uses the entire FashionMNIST and MNIST datasets (60,000 training set and 10,000 test set images each) and a subset of 70,000 NotMNIST images  to train their learners. First a multi-task classifier is trained on different combinations of datasets (MNIST + FashionMNIST, FashionMNIST + Not MNIST, etc.)  for 25 epochs with a learning rate initialized to 1e-3, then another 25 epochs with an initial learning rate of 1e-5. The weights of the trained multi task learners are then used as the initial weights for single task learners that are then trained for a specific task in the same procedure with a different dataset (i.e. an MNIST single task learner trains further on the MNIST dataset).

All hyperparameters are clearly outlined, including learning rate, optimization function used, and number of epochs ran for training. The authors mention a decay rate for the learning rate but did not specify it. Furthermore they released the code used to run their experiments, and in their code they make extensive use of batch normalization and dropout in their A-CNNs, though the paper makes no mention of it. This made it more difficult to obtain their exact results, because without the code it would have been impossible to recreate their exact training procedure. We did not include batch normalization and dropout in our architecture even after this discovery, to better gauge how well the paper's results can be reproduced based strictly on what is written in it.

This team was able to closely reproduce the author’s work with a combination of C++, Python, and bash scripts, thanks to the detailed architecture and hyper parameters outlined in the paper. A C++ code preprocessed the datasets, obtained from their official websites, and saved them in a custom format as .ocv files. Python scripts built the learners using Keras Tensorflow and fit the models to the appropriate datasets using the same hyperparameters specified by the original paper. A bash script called the Pythons scripts and passed in different arguments for which datasets to train on and what to set the initial learning rate to.

The results reported by the paper seem to favor the author’s conclusion that multi-task learners are more powerful than their single  task counterparts; on the task of classifying digits from the MNIST dataset, for example, a single task learner whose weights were initialized randomly got an accuracy of 99.56\%, whereas a single task learner initialized from multi task learner trained on the MNIST and Fashion MNIST datasets got an accuracy of 99.71\%. This team obtained some results that agreed with the author's conclusion, but others that contradicted it. For the same set of comparative results, for example, this team obtained accuracies of 99.36\% and 99.21\% respectively, where the single task learner out performed its multi task counterpart.

Overall the results obtained by the authors were easy to reproduce as the authors provided ample detail of their method and architectures used. This team obtained some results that correlated with the author's conclusions, and some that did not. These discrepancies could be attributed to methods used by the others that went unmentioned in the paper, though with results between classifiers differentiating by less than half a percentile in some cases (both ours and the authors) it is difficult to discern a significant trend from regular fluctuation.

---

### Public Comment · (anonymous) · 2017-12-16
**Reproducibility Challenge**

REPLICATED MODEL

The approach presented in this paper involves the application of the multi-task learning methodology to a supervised learning task in order to determine the effectiveness of using multi-task learning versus using only single-task learning. The data-sets used included three MNIST-like data-sets: MNIST, Not-MNIST, and Fashion-MNIST.

The multi-task learning paradigm involved pre-training the network with different combinations of the three data-sets. These combinations included three single-tasks, three bi-tasks, and one tri-tasks. The single-tasks refer to simply using each of the three data-sets individually. The three bi-tasks include the three possible combinations of two data-sets: MNIST+Not-MNIST, MNIST+Fashion-MNIST, and Not-MNIST+Fashion-MNIST. The tri-task referred to using all three data-sets for pre-training.

The weights from the pre-trained networks are then used to initialize a training network for each data-set individually. Note that for the bi-task learning, the image recognition task is not performed on the data-set that was not involved in the pre-training process. For example, if the pre-training was done on a pooled Not-MNIST and Fashion-MNIST data-set, then the image recognition is not performed on the MNIST data-set as the MNIST was not involved in the pre-training process. In addition, for the single-task learning, there was no pre-training procedure.

The architecture of the neural network was the all convolutional neural network. The whole architecture was constructed using Tensorflow in the replication.

For the multi-task learning models, the neural network is first pre-trained for 50 epochs in the multi-task context (bi-task or tri-task). With the parameters transferred over, the neural network is then re-trained for another 50 epochs in the single-task context before conducting the final image-recognition, for a total on 100 epochs on the multi-task learning models. For the single-task learning, the neural network is solely trained for 50 epochs on the individual data-set before performing the image-recognition task.

For each instance of the 50 epochs, a two-stage learning rate decay scheme is employed. For the first stage of 25 epochs in the instance, the learning rate was initialized to 10^{-3} and in the second stage of 25 epochs, the learning rate initialization was reduced to 10^{-5}.

REPLICATION RESULTS

The results of the replication are as follows: On the MNIST data-set, the single-task achieved 99.59%, MNIST+Fashion-MNIST achieved 99.61%, MNIST+Not-MNIST achieved 99.58%, and MNIST+Fashion-MNIST+Not-MNIST achieved 99.64%. On Not-MNIST, single-task achieved 97.02%, Fashion-MNIST+Not-MNIST achieved 96.95%, MNIST+Not-MNIST achieved 97.08%, and MNIST+Fashion-MNIST+Not-MNIST achieved 96.91%. On Fashion-MNIST, single-task achieved 93.92%, Fashion-MNIST+Not-MNIST achieved 93.81%, MNIST+Fashion-MNIST achieved 93.56%, and MNIST+Fashion-MNIST+Not-MNIST achieved 93.57%.

The original study indicated a consistent increase in accuracy from single-task to tri-task learning, with a general pattern of increase in accuracy going from single-task to multi-task models. The results we obtained did not match this pattern in general. 6 of the 9 multi-task models had a lower accuracy than the corresponding single-task models. In addition, the single-task model achieved the highest accuracy for the Fashion-MNIST data-set.

The discrepancy in the results is based on very small absolute differences. Since each model was trained only once, the difference in results may have been simply due to chance. Keeping this in mind, further analysis using more training experiments and subsequent statistical significance analysis would be helpful for ascertaining the accuracy of multi-task learning versus single-task learning for MNIST and MNIST-like data-sets.

---

### Decision · Program_Chairs · 2018-01-29
**ICLR 2018 Conference Acceptance Decision**

**Decision:**

Reject

**Comment:**

the paper validates the benefit of multi-task learning on MNIST datasets, which is not sufficient for ICLR publication